# Millennials' Awareness and Approach to Social Responsibility and Investment—Case Study of the Czech Republic

**Sylvie Formánková [1,\*], Oldřich Trenz [2], Oldřich Faldík [2], Jan Kolomazník [2] and Jitka Sládková [3]**

[1] Department of Management, Faculty of Business and Economics, Mendel University in Brno, 613 00 Brno, Czech Republic

[2] Department of Informatics, Faculty of Business and Economics, Mendel University in Brno, 631 00 Brno, Czech Republic; oldrich.trenz@mendelu.cz (O.T.); oldrich.faldik@mendelu.cz (O.F.); jan.kolomaznik@mendelu.cz (J.K.)

[3] Department of Economics and Management, Sting Academy, 637 00 Brno, Czech Republic; 10282@post.sting.cz

[\*] Correspondence: sylvie.formankoval@mendelu.cz; Tel.: +420-777-160519

**Abstract:** We live in a new era with contradictory views on society, industries, and the whole world. Raising consumerism is compensated by raising the responsibility of the new generation, the so-called Millennials. The paper deals with the awareness of Millennials of corporate social responsibility (CSR), and their attitude to sustainable and responsible investment (SRI). The research is focused specifically on students of economically oriented higher education institutions (HEI), studying in the Czech Republic. For the purpose of general awareness of the term CSR, a sample of 1073 HEI students from different institutions was analyzed and evaluated. For the evaluation of their attitude to SRI, 213 respondents from Mendel University in Brno were interviewed. The research brought surprising results—bachelor's degree students have a better awareness of the term than master's degree students. This surprising fact can be explained by the fact that CSR courses have been incorporated into studies in recent years. Another important finding from the second research is that 57% of respondents are willing to sacrifice part of their return in the case of an investment in socially responsible instruments. This fact can be used for the design of an investment strategy offered by investment institutions.

**Keywords:** corporate social responsibility; Millennials; sustainable and responsible investment; students

## 1. Introduction

The awareness and development of the corporate social responsibility (CSR) concept as a management practice started in the United States (U.S.) in 1950–1960 [1]. In the Czech Republic, authors began talking about this topic in the second half of the 1990s [2]. Currently, we live in a new era with contradictory views on the society, industries, and the whole world. Raising consumerism is compensated by raising the responsibility of the new generation, the so-called Millennials. Corporate social responsibility is an integral part of their life. Gen Y individuals (another expression for Millennials) seek to balance "quality of life" and the "quest for wealth" [3]. Many authors believe this generation to be more aware of corporate social responsibility. However, is that so in all aspects of their life, in all countries, or only in the U.S.? How about their own behavior? Are they willing to sacrifice their utility for a good cause? This trend and behavior can be observed in the financial industry. Morgan Stanley, the American multinational investment bank and financial services company, announced following statement in the year 2017: "A younger generation of investors, who overwhelmingly

believe that their investment decisions can make an impact, is leading the sustainable investing charge." More investors are interested in sustainable investing and adopting its principles as part of their strategy, but Millennials are leading the charge, according to the Institute for Sustainable Investing's 2017 "Sustainable Signals" report. Among individual investors, 75% are interested in sustainable investing. A total of 86% of Millennials are interested. Millennial investors are making more sustainable investing decisions, believe that their investment can create positive change, and want more proof of performance, but remain committed to sustainable investing [4–6]. Since Millennials are poised to receive more than $30 trillion of inheritable wealth, sustainable investments will continue to grow in demand. As a result, fund managers are increasingly allocating resources to develop products, and to capture this emerging client segment. Financial planners need to refresh the way in which they approach Millennial prospective clients [6,7]. Generation Y will become an increasingly influential consumer segment during the next decade, meaning that looking at the images of the future of the Millennials and the views that the currents middle-aged generation have on the Millennials future values and lifestyle, reflected in their investment behavior are extremely relevant for the financial business [8].

Despite the above explained importance of the upcoming generation of Millennials, when examining the results from the Web of Science database, only a few issues have dealt with the Millennial approach to social responsibility or investing in the last five years. Table 1 shows the gap in the CSR literature resources, scientific papers, and studies:

**Table 1.** Web of Science database search (last five years), dated to 8 January 2019.

| Key Words | Number of Issues Found | Relevant | Irrelevant | Number of Issues Written in the Czech Republic |
|---|---|---|---|---|
| Millennial | 2083 | 452 | 1631 | 0 |
| Millennial + behavior | 192 | 98 | 94 | 0 |
| Millennial + investing | 7 | 1 | 6 | 0 |
| Millennial + investment | 12 | 2 | 10 | 0 |
| Millennial + social responsibility | 21 | 5 | 16 | 0 |
| Millennial + CSR | 8 | 5 | 3 | 0 |
| Millennial + Czech Republic | 2 | 0 | 2 | 0 |
| Generation Y + social responsibility | 21 | 4 | 17 | 1 |
| Generation Y + invest | 7 | 2 | 5 | 0 |
| Generation Y + CSR | 6 | 1 | 5 | 1 |
| Generation Y + Czech Republic | 21 | 8 | 13 | 20 |

Source: Own elaboration based on searching in the Web of Science database [9].

The relevant issues are those papers that deal with the same meaning of the key words or their combination. The irrelevant issues do not deal with the same meaning as the searched key words. Looking closer at the results, there were only 98 relevant papers studying the behavior of Millennials, and only a few papers investigating them in the relation to social responsibility or investment. We decided to search the key word Generation Y as well, since authors use the terms Generation Y and Millennials interchangeably (as explained later in this paper). None of the papers were written in the Czech Republic. In the Web of Science analysis, we found out that the majority of the papers were written in the U.S., and most of them deal with Millennial values, and factors influencing them as job seekers [10–12] or customers [13–15]. The topic of studying the Millennial generation, and their approach to social responsibility and investing, has not been examined in appreciated scientific sources yet, and it opens the space for further investigation.

Therefore, the aim of this paper is to examine the awareness of the Czech Millennials towards CSR, and their attitude towards sustainable and responsible investment (SRI). We base these hypotheses on the premises resulting from the literature review. Since there have not been any studies dealing with this issue among higher education students in the Czech Republic, we focus specifically on students

of economically oriented higher education institutions (HEI), studying bachelor, master, and Ph.D. studies in the Czech Republic. We selected this group of students because they create a huge part of chief executive officers, leaders and potential investors in the future (as explained later in the text), and they have the opportunity to attend courses that are related to CSR. In the future, we will extend this research to students of other fields, such as technical, medicine, etc.

The results of our research will help us to design, implement, and verify the model and methodology of sustainable investment (SI), taking into account ESG (economic, social, and governance) factors in selected sectors providing investors with a framework for SI evaluation, using advanced maths, statistical, and econometric methods, and the Web portal. The model can be used by teachers in their CSR tuition, by researchers for data mining and evaluation, for investors to help them with their decision making, and for investment companies, to help them to understand the behavior and approach to CSR and SRI of the upcoming generation of investors.

## 2. Theoretical Background

### 2.1. Corporate Social Responsibility vs Sustainable and Responsible Investing

Almost everyone today, whether students, managers, or employees, has heard something about the notion of CSR, whether in the business community, from the media, or even within popular culture [16]. Harold Bowen is considered the founder of CSR, defining CSR in 1953 that "refers to the obligations of businessmen to pursue policies, to make those decisions, or to follow those lines of action which are desirable in terms of the objectives and values of our society" [17]. The Commission puts forward a definition of CSR as "the responsibility of enterprises for their impacts on society" [18,19]. The Strategic document, the National Action Plan of Corporate Social Responsibility in the Czech Republic (NAP), provides this definition of CSR: "In the current approach CSR represents a coherent set of activities and practices that are an integral part of the control strategy of the organization in the social, environmental and economic area, the organization provided them beyond the legal obligations and with motivation to contribute to the improvement of conditions in the society" [20,21]. Nevertheless, a universal definition remains elusive. There is a large number and diversity of the approaches to determine the CSR concept [22–24]. According to Cole, this can be largely attributed to the fact that the concept has scope to mean different things to different people in different circumstances [25]. Notwithstanding this, all firms engaging with CSR are essentially pledging to conduct their business in ways that protect the interests of current and future generations, and are trying to eliminate or minimize any harmful effects and maximize its sustainable beneficial impact on society (Mohr et.al. 2001 as cited in [26]). CSR overlaps with other concepts, including corporate citizenship, corporate sustainability, environmental management, sustainable development, triple bottom line, and more recently, with "shared value". These terms are often used interchangeably, despite the continuing debate on differentiating the terms (Porter and Kramer, 2011 as cited in [27]; Montiel, 2008 as cited in [28]). It is also part of the organizational culture [29,30].

The role and importance of CSR and business ethics is especially evident in controversial sectors of the economy, such as the pharmaceutical, tobacco, alcohol, and mining industries [31–36]. The financial industry might be identified as one of these sectors as well. The corporate social responsibility starts to appear more and more frequently in the area of investment. An unequal definition of what belongs under ethical investing, and what is out of its scope, is a major drawback of SRI. The parameters of ethical investing are not kept in ledgers, and are therefore very subjective [37]. One of the definitions of SRI (socially responsible investment in USA/ethical investment in Europe) is according to Michelson at al.: "…the integration of personal values, social considerations and economic factors into the investment decision." [38]. SRI is another concept that is problematic, due to its lack of a consistent terminology [39]. SRI encompasses different styles of investment decisions and investor behavior, such as "ethical investing", "value-based investing", "cleantech investing", "impact investing", "sustainable investing", and simply "responsible investing" [1]. European SRI Study introduces the classification

of sustainable investment strategies [40]. The Forum for Sustainable and Responsible Investment broadly defines SRI as an investment process that integrates environmental, social, and governance considerations into investment decision making to generate long-term competitive financial returns and positive societal impact. It is a process of identifying and investing in companies that meet certain standards of Corporate Social Responsibility [41]. We will operate with the following definition: "Sustainable and responsible investing (SRI) is an investment policy that adds a third dimension to the risk to investment return ratio, namely the social responsibility."

SRI industry has experienced rapid growth in recent times. The importance of the SRI industry is growing fast, and it is becoming a phenomenon that has to be taken seriously into account by both researchers and business experts. The number of investment funds that apply a social/environmental screen to company stocks is also growing rapidly [42–45]. According to Donovan, we are now living in the SRI 2.0 period, which is focused mainly on impacted investment and the extension of the general awareness of responsible investing [46]. However, the idea of socially responsible investment (SRI) is not a matter of recent years. For example, in the 18th century, members of some churches refused to invest into companies whose activities on society had at least a controversial impact [47]. Furthermore, the idea of socially responsible investing was highlighted, for example, after the Civil War in the United States, when society was looking for new ways to change the system, which was unable to provide enough assistance to those in need in a way other than through charity. Socially responsible principles were promoted by such figures as Andrew Carnegie and John D. Rockefeller [48]. On the main global stock exchange markets, social and ecological business responsibility indexes have been functioning for years. The American Dow Jones Sustainability Indexes (DJSI) and Calvert Social Index (CSI), English/London FTSE4Good Index Series, FTSE JSE Johannesburg Stock Exchange Responsible Investment Index (RPA), Brazilian Sao Paolo Stock Exchange Corporate Sustainability Index (ISE), Indian Standard & Poor's India Environmental, Social and Governance Index (S & P ESG). In Poland, the RESPECT Index has been listed as the first one in the Central and Eastern Europe. [49] In the Czech Republic, currently, Česká spořitelna (Czech Savings Bank) is the first institution offering SRI funds. In 2018, the first fund called Tilia Impact Ventures, focused on companies with social impact, was founded [5–51]. Besides SRI funds, the CII750 index tracks the performance of 750 investment funds operating on the Czech capital market [52].

## 2.2. Millennials and Their Approach to Sustainable and Responsible Investing

Financial decisions are made based on both cognitive and affective dimensions. This means that personal values, emotions, personality traits and societal influence financial decision making, especially fear and greed fuel bubbles in the financial markets. (Olsen, 2010 and Landberg, 2003 as cited in [8]) However, investment decision-making is based primarily on the risk-benefit comparison, with SRI being no exception. Although the study by Professor of Corporate Finance at Tilburg University, Luc Renneboog, Finance Professor at Tias Nimbas Business School, Jenketer Horst, and Professor of Finance at Warwick Business School, Chendi Zhang, published in 2008 in the Journal of Banking & Finance, comes to the conclusion that investors within SRI are willing to accept lower returns in exchange for the socially responsible investment factor [53]. This is what the American hedge fund manager, Cliff Asness [37], points out; the former thinks that SRI is a loss matter, where the investor pays one part of its profit for investment responsibility. The founder of the Conscious Capital advisory firm, Derek Tharp, also points out that people who consider socially responsible investing should be prepared for the fact that their contribution to sustainable development might equal zero [54]. However, studies focused on the comparison of SRI performance and risk rate largely agree that the impact of ESG on investing is non-negative. Investors can thus invest with similar returns and risks, but into socially responsible companies [37]. An analysis of the Nuveen company of July 2017 shows that no statistically significant difference in return was found, compared to the general market benchmarks, when the most significant SRI stock indices were subject to valuation. In the long run, therefore, there was no performance penalty for socially responsible investing [54]. Trnková (2004)

also states that investing into socially responsible companies is considered less risky with a possible above-average return [55]. Some studies have found no statistically significant relationship between returns of SRI funds and conventional funds [56]. The picture is complex, and an analysis may be different according to the type of methodology or screening involved [39]. Regardless of any of these opinions, the investor always has to answer one important question: Am I willing to sacrifice a part of my risk, knowing that the fund is sustainable and that the companies are responsible to society and the environment? The new generation of investors seems to be more open to thinking about these questions. This new generation is called the Millennials.

Authors have different attitudes towards the definition of Millennials. Some believe that people born in or after 1982 belong in this group; some say Millennials were born between 1980 and 2000. According to David Foot, Millennials are people born between 1980 and 1995. He also refers to them as the "Baby Boom Echo", since Millennials are the children of the Baby Boomers (1946–1965) [57]. They are called by different names; apart from Millennials, we can read about the Nexters or Generation Y [58]. Nevertheless, these authors often share an opinion on their behavior, and people that belong to Generation Y are considered to share similar consumption patterns and culture [59]. "The age group that shows the deepest sensitivity toward ethical and CSR issues is comprised of young people, born in or after 1982, often recognized as Generation Y or the Millennial Generation (Howe & Strauss, 2000; Cone Communications, 2006; Ciemniewski & Buszko, 2009; Connell et al., 2012). The so-called Millennials share a widespread belief that their responsibility is to make the world a better place (Cone & AMP Agency, 2006)." [11] Millennials may face three major problems in relation to management learning: lack of concentration, lack of engagement, and lack of socialization [60]. On the other hand, they have a new way of thinking, including a lot of information gained online, and they select the information that they consider important. Millennials are also receptive and open to trying alternative means of ownership. Millennials have: "...more concern for others and less interest in material goods [...] [are] less interested in keeping up with materialistic trends and less invested in obsessive consumerism as a way of life." (Rifkin, 2014, p. 224 as cited in [61]). They tend to be individualistic, and they make decisions based only on a few factors or attributes that are important to them, such as efficiency, timing, and mediums of communication (Landberg, 2003 as cited in [8]). However, as well as other people, they are influenced by the cultural factors such as factors defined by Geert Hofstede: power distance, individualism vs collectivism, masculinity vs femininity, uncertainty avoidance, long term vs short term orientation, and indulgence vs restraint [62]. It is hypothesized that the values of U.S. millennials would be most similar to those of millennials in countries with a national culture much like the U.S., and dissimilar to those of millennials in countries with a national culture differing from that of the U.S., as evaluated by the Hofstede model on national culture. The results of the research by Schewe et.al [63] are that it would be wise to consider each country's scores on the Hofstede dimensions when initially evaluating the similarity of millennials in different countries. They found a greater similarity in the values of millennials when a country was most closely aligned on the Hofstede dimensions. Another factor influencing Millennials in their decision making is the history of their country of origin and the level of CSR development there. In its most well-known guise, CSR is essentially a U.S. idea. It was in the U.S. where the language and practice of CSR first emerged. In other parts of the world, most notably Europe, the Far East, and Australasia, however, there has always been a stronger tendency to address social issues through governmental policies and collective action. Between those two major categories of developed and developing countries, there is a third category that deserves attention from a CSR perspective. Most countries of the former communist bloc have changed from a planned and government run economy to a capitalist market system [64]. Here belongs the Czech Republic. The development of CSR, therefore, started later in this country than in the U.S. or other developed countries which has to be taken into consideration [1,2]. Except the general factors, a good place to start in examining the individual influences on ethical decision-making is to consider some basic demographic factors, such as gender and age. For example, one common question is whether men or women are more ethical. However, the overall results have been less than conclusive, with

different studies offering contradictory results, and often no differences found at all. A similar problem is present with age as with gender [65]. According to Lawrence Kohlberg, the difference is not in our age, but in our stage of cognitive moral development [66]. He introduced three broad levels of moral development: Level one—the individual exhibits a concern with self-interest and external rewards and punishment; level two—the individual does what is expected of them by others; level three—the individual is developing more autonomous decision-making based on principles of rights and justice, rather than on external influences. The Trevino's person–situation interactionist model combines individual variables (moral development, etc.) with situational variables to explain and predict the ethical decision-making behaviors of individuals [67]. However, factors leading investors (regardless of their generation) to choose SRI products are not still well understood [68]. According to Credit Suisse, Morgan Stanley, and Ernst & Young, Millennials, as the next generation of investors, have a clear vision when it comes to investments. A major factor that unites this generation and that differentiates them from the previous generations is a sense of global connectivity, combined with a feeling of collective responsibility for the well-being of the world. In terms of investments, this distinct millennial philosophy means that they do not just care about financial returns. Millennials want to see what impact their investments have, and how they can do good for society or the environment [4–6,69].

## 3. Materials and Methods

### 3.1. Research Methodology No.1: The Awareness of the Term CSR

#### 3.1.1. The Sample

The research was realized by the authors of this paper at the Faculty of Business and Economics, Mendel University in Brno. The research was conducted among Millennials, namely, students of economically oriented higher education institutions (HEI) in the Czech Republic (CR). Before the investigation of the Millennials' attitude to sustainable investing, it is important to know their awareness and knowledge of the term CSR. For this purpose, a sample of 1073 HEI students from 14 different higher education institutions in CR was analyzed and evaluated (from a total of 48 HEI with economic fields of studies). In this quantitative research, the form of half-structured questionnaires was chosen and distributed online (a preview of the questionnaire is available here: https://umbrela.mendelu.cz/research/1640/preview#/). We used the university system for questionnaire creation and elaboration available here: https://umbrela.mendelu.cz [70].

#### 3.1.2. Data Collection and Analysis

The first collecting period was from September 2017 to November 2017, and the second round of data collecting was from December 2017 to February 2018. In our research, we decided to confirm the validity of the premises from the literature review, pointing that general knowledge of the term CSR among Millennials was good, and that we wanted to find out if the level of education played a role in this fact. We asked following research questions (RQ):

RQ1: Are the students of economically oriented high education institutions familiar with the term CSR and do they know exactly what it means?
RQ2: Does the level of education influence the awareness of CSR?

Universities plays a fundamental role in CSR education since they are the greatest contributors to the formation of their students, forthcoming entrepreneurs, business leaders, managers, and employees. Recently, students majoring in business administration have been exposed to the concept of CSR in a number of courses [71]. For this reason, and based on our experience and current study plans, we assumed that the higher the level of education of students, the better the awareness of the term CSR. We set the hypothesis $H_0$: "The higher education level (bachelor, master, Ph.D.) achieved by the student, the better the awareness of the term CSR". We tested the awareness of this term according to the level of studies (bachelor, master, Ph.D.).

The hypothesis was tested using the Plotly library used for the multilevel program language Python, available here: https://plot.ly/python/t-test/ [72]. The results were also tested by another statistical program (software) called Statistica (http://www.statistica.pro) [73]. The data were verified on two independent platforms (Plotly, Statistica). A *t*-test with a level of significance (*p* value) of 0.05 was applied. The *t*-test (also called Student's *T* Test) compares two averages (means) and tells us if they differ from each other. The *t*-test also presents how significant the differences are [74].

$$t = \frac{\mu_A - \mu_B}{\sqrt{\left[\frac{\left(\sum A^2 - \frac{(\sum A)^2}{n_A}\right) + \left(\sum B^2 - \frac{(\sum B)^2}{n_B}\right)}{n_A + n_B - 2}\right] \cdot \left[\frac{1}{n_A} + \frac{1}{n_B}\right]}} \tag{1}$$

where:

$(\Sigma A)^2$: Sum of data set A, squared,
$(\Sigma B)^2$: Sum of data set B, squared,
$\mu_A$: Mean of data set A,
$\mu_B$: Mean of data set B,
$\Sigma_A{}^2$: Sum of the squares of data set A,
$\Sigma_B{}^2$: Sum of the squares of data set B,
$n_A$: Number of items in data set A,
$n_B$: Number of items in data set B.

### 3.2. Research Methodology No.2: The Attitude of Millennials to Sustainable and Responsible Investing

#### 3.2.1. The Sample

The second area of interest was the attitudes of Millennials to sustainable and responsible investing. In this case, we started with a study focused on the students of the Faculty of Business and Economics, Mendel University in Brno. Here, we achieved 213 respondents. More than one half of respondents already had jobs (51.1%).

#### 3.2.2. Data Collection and Analysis

Data collection ran from February to May 2018. The research was performed in the form of an online questionnaire survey (see Attachment No.1). The preview of the questionnaire is available here: https://umbrela.mendelu.cz/research/1612/preview#/. The following research questions (RQ) were investigated:

RQ1: Whether or not the respondents have already considered investing their funds.

RQ2: What criteria for investment decision-making are crucial for them.

RQ3: To what extent is it important for them, when making investment decisions.

RQ4: Whether or not the instrument in which the money is invested (fund, etc.) is sustainable and socially responsible.

RQ5: Whether or not they are willing to risk more in order to achieve a higher return.

RQ6: Whether or not they are willing to sacrifice part of the return if it is an investment in socially responsible instruments.

Due to the facts mentioned in the literature review regarding many factors influencing Millennial decision making (cultural, historical, individual, etc.) we believe that the factor of having a job does not affect their willingness to sacrifice part of their return, in case they are investing in SRI funds. Here we set the null hypothesis $H_0$: "There is no difference in the willingness to sacrifice part of the return

in the case of an investment in socially responsible instruments between working and non-working students".

The hypothesis was tested with the *t*-test, using a *p*-value of 0.099. This part of research will be later extended on the whole Czech Republic, and the aim is to find partners abroad to be able to compare the data in different countries.

The whole study was part of our project "Modelling and simulation of sustainable investment decision-making". The aim of the project is to design, implement, and verify the model and methodology of sustainable investment (SI), taking into account ESG factors in selected sectors providing investors a framework for SI evaluation with using advanced maths, statistical and econometric methods, and Web portal.

## 4. Results

### 4.1. Millennials' Awareness of Corporate Social Responsibility

According to previously mentioned authors, Millennials tend to behave more responsibly in their life, and that should be reflected in their attitude towards investment. If this is true, it is good to know whether they know what the term CSR actually means. In our sample (sample and process described in the chapter Methodology) we found that only 27% (absolute frequency = 288) of respondents thought that they knew the term CSR, and claimed that they knew exactly what it meant. Some of them had already heard of the term CSR (35%; absolute frequency of 378) but there was still a group of students (38%, an absolute frequency of 407) who had never heard of the term CSR (see Figure 1).

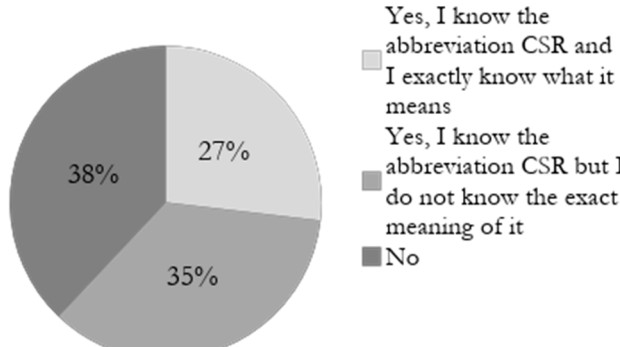

**Figure 1.** Awareness of the term CSR among Czech higher education students.

We set the hypothesis H$_0$: "The higher the education level (bachelor/master/Ph.D.) achieved by the student the better awareness of the term CSR". The results of hypothesis testing showed that students without any academic degree (bachelor/master/Ph.D.) studying the bachelor level of studies were aware of the CSR term. On the contrary, students at the master level of studies did not know CSR. Ph.D. students were already aware of this term.

We based our hypothesis on current study plans and the assumption that students already passed courses related to the CSR topic. The surprising results could be explained by the fact that CSR courses have been incorporated in studies in recent years. There has been a lack of attention to, and discussion of, CSR in the Czech Republic, and in relation to education [61]. According to Stonkute et al. the issues of business ethics, corporate social responsibility, and sustainability have attracted increased attention in management education in recent years, and a five-fold increase in the number of stand-alone ethics courses has been noted since 1988 [75]. The same situation is observed in other courses that are not inherently focused on CSR, such as marketing, public relations, accounting, etc., which have started to include CSR in their syllabuses in the last few years. Positive development in the higher level of involvement in CSR training modules has also been described by Adámek [76]. According to his research in 2013, Corporate Social Responsibility courses were offered by 21% of higher education

institutions. According to Srpová et al., in the year 2012 the amount of such research focusing on the topic of CSR was increasing, and an award exists for the best thesis in CSR, despite the fact that courses on CSR are not widely taught in Czech universities [77]. Nevertheless, the number of conferences and specialized seminars has increased recently. Thus, there is a real chance that students of master studies might possibly not have attended courses oriented towards CSR, and that could be one of the reasons why they do not know this term. A second reason might be that students had heard about the term during their bachelor studies, but they did not remember (due to a lot of information learned during studies, focus on one direction at the master's level, etc.). Their pure knowledge is contrary to the increasing public awareness of CSR in the Czech Republic [78]. However, as stated by Berényi and Deutsch, higher education shall intensify the education of the field [79]. This is supported by the opinion of Tormo-Carbó et.al.: " . . . it is crucial to improve the effectiveness of business ethics and corporate social responsibility (CSR) education, in terms of its impact on business students' awareness of ethical issues." [80]. Support from institutions, the business sector and the government will be crucial for future success of CSR teaching [75].

*4.2. The Attitude of Millennials to Investing*

The results of our research prove that the rising generation of Millennials, in our case, Czech higher education students, already considers the possibilities of investing their funds (73.7%). This fact can be complemented by the outcomes of the research that were also conducted at the Faculty of Business and Economics, Mendel University in Brno, where it has been established that these students "would like to live in a society that is more focused on the highest economic performance, technical advantages, the decisions are taken by experts, life is driven by rules (not by freedom) and they prefer modernization changes (not the maintenance of traditions). It can be summarized that the majority of students at the FBE are technological, modernizing and expert optimists." [81].

Though the criterion of whether or not their funds are invested in socially responsible and sustainable instruments was ranked only among the last ones in investment decision-making (see Table 2), 40.8% of respondents considered this criterion to be very important (16) or somewhat important (71) when making decisions. On the contrary, this factor was not important for 38.5% of respondents. The rest did not know (20.4%). Nevertheless, expected return still remained in first place, in comparison with other criteria.

**Table 2.** Preferences of respondents in investment decision-making.

| Sequence | Criterion |
|:---:|:---:|
| 1 | Expected return |
| 2–3 | Guaranteed rate of return |
| 2–3 | Rate of loss risk |
| 4 | Return-to-loss ratio |
| 5 | Whether or not the investment is time-limited, or whether there is the option of immediate withdrawal of funds |
| 6 | Whether or not my funds are invested in investment instruments that do not affect society and the environment negatively (tobacco industry, alcohol, etc.) |
| 7 | Maximum possibility of fund control and handling ("I want to manage everything myself") |
| 8 | Whether or not my funds are invested in socially responsible and sustainable instruments (i.e., in funds preventing negative impact on society and promoting socially responsible activities) |
| 9 | Services provided by an investment intermediary ("I do not have to worry about anything") |

In order to achieve a higher return, 45.5% of respondents were willing to take a higher risk, whilst 54.5% of respondents preferred a lower return with minimized risk.

An important finding was that 57% of respondents were willing to sacrifice part of their return in the case of an investment in socially responsible instruments (see Figure 2). It can therefore be assumed that, in the case of the selection of one's own investment instrument, the expected return would still prevail over sustainability and social responsibility; however, if the SRI investment option was offered, for example, by an investment adviser, the majority of respondents would be willing to use this investment instrument, even at the expense of a lower return. However, we can state that the respondents would care about the negative impact of their investment prior to the investment with positive impact on the society. That means that they would care if their investment did not cause any harm, and only then would they care if their investment also contributed to the prevention of negative impact.

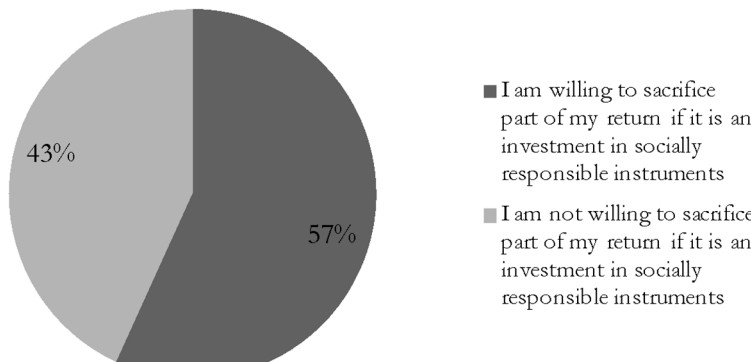

**Figure 2.** Respondents' willingness to sacrifice part of their return in the case of an investment in socially responsible instruments.

The Millennial generation are increasingly becoming important actors in business, and by 2025, they will make up 75% of the workforce (Deloitte 2014 as cited in [76]). A total of 56% of Millennials were likely to refuse to work for a company after learning that it was not socially or environmentally responsible [82,83]. They need recognition, an ensured work–life balance, professional development, and other attributes of a responsible company/management attitude [84]. They want to see impacts. This fact has been confirmed by Petr Šmíd, the head of marketing department in Google Czech Republic [85]. CSR is key here for both enhancing the view of an organization by its current employees, as well as attracting the best young talent [82]. While Millennials are commonly criticized for lacking in work experience, having a weaker work ethic, and being entitled, a growing number of them are already acquiring work experience during their studies. (Twenge 2010; Ernst & Young 2014 as cited in [82]). When observing the differences among the students in our sample, those who were already working and those who were not working, here, the defined hypothesis was $H_0$: "There is no difference in the willingness to sacrifice part of the return in the case of an investment in socially responsible instruments between working and non-working students". In our case, this hypothesis was accepted (Table 3).

**Table 3.** Test of hypothesis.

| $H_0$ | *t*-test | Statistic = $-1.6528$ | *p*-Value = 0.099 |
|---|---|---|---|

## 5. Discussion

The aim of this paper was to examine the awareness of a special group of Millennials, students of economically oriented higher education institutions, of corporate social responsibility (CSR), and their attitude to sustainable and responsible investment (SRI). The research was focused on students studying bachelor, master, and Ph.D. studies in the Czech Republic. The first part of the research studied the Millennials' awareness of CSR. The interest in the Millennial generation has increased all

around the world [86]. Some papers discuss similar issues to ours. A similar research was conducted, e.g., by Berényi and Deutsch, among Hungarian students [79]. The authors did run a survey in 2014 and in 2016 about business students' perceptions and attitudes to sustainable development and CSR. Their sample consisted of business students in Hungarian Higher education (various universities, n = 100 from each year, random sampling). The results show a slightly increasing confidence of CSR, and the attitudes of females and males seem to converge, and respondents with superficial knowledge about CSR showed an increasing level confidence in its usefulness. In conclusion, there are significant differences between the years by gender and the level of CSR knowledge, but the pattern is scattered, and the confidence in CSR did not increase clearly. A totally different set of results are observed in developing countries, such as Ghana, where a low level of CSR awareness among university students was observed [71]. Primary data collected via a survey of González-Rodríguez et al. in Business Schools at Universities of Spain, Poland, and Bulgaria, and multivariate analysis evidence that both values structures and university students' CSR perceptions are influenced by cross-cultural factors. The results indicate that students from Poland have higher perceptions about CSR, compared to Bulgarian and Spanish students. This comparison and its conclusions explain the country differences in this High Education context and according to Schwartz's values theory and the Triple Bottom line, the study reveals different value profiles by gender and nationality, and diverse attitudes to CSR perception across European countries. As mentioned in the literature review, respondents from different countries are influenced by the cultural factors, their individual cognitive moral development and education, and by the historical development of the country from which they come from [62,65,66,76]. Corporate social responsibility has evolved differently not only between Western and Eastern European counterparts, but also within those regions. This is the reason for why we intend to extend our research and plan investigations in other countries [87]. One of the studies, investigating 7700 Millennials from 29 countries around the globe, was presented by Deloitte in 2016, and they are continuing their research [88]. These results might be used for the next evaluation of our research.

In terms of the Millennials' attitude to sustainable investing, it is very difficult to compare the data. There has not been a common scientific study that has been conducted in the Czech Republic, nor in the rest of the world. There are only studies developed by different institutions. One of them is Morgan Stanley. According to the survey by Morgan Stanley [4,5], 86% of Millennials were interested in sustainable investing. In our research, we found out that only 40.8% of respondents considered the criterion of whether or not their funds were invested in socially responsible and sustainable instruments that were very or somewhat important when making decisions. On the other hand, 57% of respondents were willing to sacrifice part of their return in the case of an investment in socially responsible instruments. We may expect that Czech Millennials will achieve higher numbers in the upcoming years. As mentioned earlier in this paper, the development of CSR and the awareness is slower or rather "shifted" in the comparison to the United States. Larson et al. present in their paper that "the connection between knowledge, risk, and investment is especially relevant to the millennial cohort, considering that high risk aversion (Debevec et al., 2013) and low levels of financial knowledge (Lusardi, Mitchell, and Curto, 2010) are recognized as defining attributes of that generation." [89]. This is why we took the factor of risk into consideration, and found out that in order to achieve a higher return, 45.5% of respondents are willing to take a higher risk, whilst 54.5% of respondents prefer a lower return with minimized risk. This confirms the risk aversion of millennials mentioned by Debevec et.al. [90]. Dorsainvil studied Generation Y in the United States as the new financial advisors and their clients, and the relationships between them; the paper suggests that creative innovation and technology is an important factor for keeping Millennial clients satisfied (Dorsainvil (2015) as cited in [8]). However, as mentioned before, we cannot see all millennials as the same. In the paper by Kontio et al. the factors influencing Mexican Millennials´ investment decisions are different, as well as their behavior. In this single country, they identified four images of the future of Mexican Millennials, and their approach to money and investing. Following the North Americans (strong loyalty to family,

community and investment in the long-term, for retirement, and for their children's college), Mudding through (be taken care by family, easy credit, diversification of debt), La Telenovela (easy money and power friends, money is to be spent, high-risk investments) and Going European (short-term savings, money is used to improve the quality of life) [8]. It is obvious that we need to approach Millennial investors differently according to their origins (country), cultures, personal values, and moral development.

During the research we faced several challenges and found certain limitations. As Williamson and Johanson mention in their book, there are disadvantages to using self-administrative questionnaires, especially major surveys, such as: 1. an adequate response rate is difficult to obtain; 2. representative responses from all groups in the population are difficult to obtain; 3. responders are unable to qualify answers or seek clarification; 4. complex questions cannot be asked; 5. supplementary observations are not available; 6. there is a lack of control over how and when the questionnaire is answered [91]. In our case, numbers 1 and 2 were the most relevant problems. Number one: The second part of our research was conducted in the form of a case study, which was tested on a smaller group of students—students of the FBE, Mendel University. This study will be in the next step of our research to extend to students from the entire country, and potentially from other countries. Number two: Men and women are not equally represented, which is reasoned by the fact that more women study economics and management. Another limit is, of course, is that we focused mainly on one particular group of Millennials—HEI students with an economic focus. The awareness of CSR and the attitude to investing would probably differ in the case of a group of non-students, or students of other disciplines, such as Arts, Medicine, Information Technologies (IT), etc. However, this research is part of a broader research conducted at the Faculty of Business and Economics, Mendel University in Brno, so a further extension of these results is therefore expected.

According to Gulavani et al., modern public universities should be aware of the real necessity of updating any educational program according to society´s requirements [92]. The authors of this paper involve the results of their research in their teaching (in the course Corporate Social Responsibility, where SRI is part of the lectures). As mentioned by Viederman, the challenge to business schools and others is to address the real world of institutional investing. We need a better understanding the circumstances that will encourage the broad range of institutional investors to consider and adopt sustainable investing [93]. We believe that the connection of the research, teaching, and practicing is the best way to show the real picture to the next generation of investors.

## 6. Conclusions

The results of this research helps us to design, implement, and verify the model and methodology of sustainable investment (SI) by providing investors a framework for SI evaluation by using advanced maths, statistical, and econometric methods, and the Web portal. Before the applications, it is inevitable to know the awareness and approaches of the future investors (Millennials) to corporate social responsibility and sustainable and responsible investing.

Despite the increasing public awareness of CSR in the Czech Republic, our results show that only 27% of respondents thought that they knew the term CSR and claimed that they knew exactly what it meant. Hypothesis testing brought another surprising fact, that students without any academic degree (bachelor/master/Ph.D.) studying the bachelor level of studies are aware of the CSR term. In the contrary, students on the master level of studies do not know about CSR. This fact might be explained by the fact that CSR courses have only been incorporated in studies in recent years. A second reason might be that students had heard about the term during their bachelor studies, but they did not remember it anymore. Pure knowledge and awareness of the CSR term is a challenge for HEI teachers to better implement social responsibility in specially oriented courses, as well as in all courses where social responsibility and ethics play roles (Management, Business, Economics, Marketing, etc.). It is not sufficient to implement CSR as the one and only course. As mentioned before, Millennials may face three major problems in relation to management learning: lack of concentration, lack of engagement,

and lack of socialization. Therefore, it is important to: 1. involve them in CSR activities, 2. engage them in CSR projects, 3. make them interact with the social environment, and 4. enable them to create an impact on society.

On a small sample of Millennials (students of FBE, Mendel University in Brno), we investigated their consideration of investment possibilities. 73.7% of them already considered the possibility of investing their funds. Here, we tried to find out whether they thought about the negative/positive consequences of such investments. A total of 40.8% of respondents consider the criterion of social responsibility and sustainability to be very important. Nevertheless, the expected return still remains in first place, in comparison with other criteria. An important finding, even on such a small sample, is that 57% of respondents are willing to sacrifice part of their return in the case of an investment in socially responsible instruments. Whether they already work or not during their studies does not have any influence on their opinion. These results are very important for companies focusing on investments to be able to offer funds reflecting the clients´ requirements, and to find the best way of communication with them. Millennials have a new way of thinking, including a lot of information gained online, and they select the information that they consider important. They make decisions based only on a few factors or attributes that are important to them, such as efficiency, timing, and mediums of communication. It is therefore important to choose a very clear and simple way of sharing knowledge, and to practice it with them.

To be able to see the changing behavior of this Millennial generation, and their awareness and attitudes to CSR, it is important to repeat the research every year, observe how the results change in time, and extend the realization of the research to more countries in the world. However, in the next study, researchers also have to start investigating the next generation coming after the Millennials, the Generation Z, or the so-called "Snow flake" generation, whose behavior is even more different to the behavior of Generation Y (Millennials) or Generation X (the generation before the Millennials) [94].

**Author Contributions:** These authors contributed equally to this paper. Conceptualization, methodology, investigation, writing, S.F.; Resources, data curation, O.T.; Stastistics and hypothesis testing, software elaboration, O.F.; Stastistics and hypothesis testing, software elaboration, J.K.; Resources and Discussion, J.S.

**Funding:** This research was funded by the GRANT AGENCY OF THE CZECH REPUBLIC (GAČR), project: "Modelling and simulation of sustainable investment decision-making", grant number CEP ID GA17-23448S".

**Conflicts of Interest:** The authors declare no conflict of interest.

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
