# Peer review of "Millennials’ Awareness and Approach to Social Responsibility and Investment—Case Study of the Czech Republic"

_sustainability, doi:10.3390/su11020504_

Reviewer 1 Report

Review Report

According to authors, the study aims to “examine the awareness of the Millennials of corporate social responsibility (CSR) and their attitude to sustainable and responsible investment (SRI)”.

Considering that Millennials are increasingly becoming important actors in business, I firmly believe that there is a real need to conduct researches focusing Millennials’ attitudes and behaviours towards CSR issues, addressing thus a significant gap in the CSR literature.

Yet, unfortunately, although in general the study is well presented, I don’t think that the paper rises to the standards expected by the Sustainability journal. While the issues approached could be interesting, for a journal focused on sustainability issues, unfortunately, I have several concerns and questions that, as a reviewer I would like to bring up:

1.     Overall, it is unclear what is really new in this paper from a CSR theory or managerial practice perspective.

2.     Authors have failed to show a significant gap related to the studies examining Millennials’ attitudes and behaviours towards CSR issues, and consequently in justifying properly the paper’s purpose. The introduction fails clearly in explaining the real significance of the study. Indeed, authors refer that “The aim of this paper is to examine the awareness of the Millennials of corporate social responsibility (CSR) and their attitude to sustainable and responsible investment (SRI)”. Why is it important? What is authors’ purposes behind this objective?

3.     Lack of significant key literature (published in relevant journals) concerning Millennials’ attitudes and behaviours. Splitting the Introduction and Theoretical Background section into two different sections would turn this evident.

4.     Hypotheses proposed lack sufficient theoretical support. Moreover, there is not a clear match between research questions and hypotheses. For example:

RQ1: “Does the level of education influence the awareness of CSR?

Based on literature authors assume that “the higher the level of education of students, the better the awareness of the term CSR”, and, in accordance, set the hypothesis:

H0: “Students of economically oriented high education institutions are familiar with the term CSR and know exactly what it means.” 

It is clear that the hypothesis H0 proposed doesn’t reflects the influence of the level of education of students, and thus doesn’t contribute to the research question …

5.     Authors should provide a better results discussion. focusing on how they can be interpreted comparing to previous studies. In fact, the discussion section corresponds to a set of general ideas, and a presentation of few previous studies, conducted in other contexts, but without a real comparison/discussion of results.

6.     The Conclusions section is fact a repetition/summarizing of results, failing in explaining the different contributions of the study.

Nevertheless, although, in my opinion, the paper lacks significant contributions to be published in this journal, I recognize that the paper is reasonably well presented and may be of interest for other journals.

Author Response

Dear Reviewer, 

thank you for your comments. We accepted all of them and reacted according to your recommendation. Here are our answers:

1.     Overall, it is unclear what is really new in this paper from a CSR theory or managerial practice perspective.

2.     Authors have failed to show a significant gap related to the studies examining Millennials’ attitudes and behaviours towards CSR issues, and consequently in justifying properly the paper’s purpose. The introduction fails clearly in explaining the real significance of the study. Indeed, authors refer that “The aim of this paper is to examine the awareness of the Millennials of corporate social responsibility (CSR) and their attitude to sustainable and responsible investment (SRI)”. Why is it important? What is authors’ purposes behind this objective?

1+2 - We explained the gap and the importance of this research. I believe it is clear now. We completed the text with other articles pointing the importance of this issue.

3.     Lack of significant key literature (published in relevant journals) concerning Millennials’ attitudes and behaviours. Splitting the Introduction and Theoretical Background section into two different sections would turn this evident.

We completed the paper with other sources published in relevant journals and other. We split the Introduction and Theoretical Background. We followed the template, that´s why it wasn´t done before.

4.     Hypotheses proposed lack sufficient theoretical support. Moreover, there is not a clear match between research questions and hypotheses. For example:

RQ1: “Does the level of education influence the awareness of CSR? 

Based on literature authors assume that “the higher the level of education of students, the better the awareness of the term CSR”, and, in accordance, set the hypothesis:

H0: “Students of economically oriented high education institutions are familiar with the term CSR and know exactly what it means.” 

It is clear that the hypothesis H0 proposed doesn’t reflects the influence of the level of education of students, and thus doesn’t contribute to the research question …

 I totally agree - I corrected it. It didn´t have any influence on the data elaboration. We also used one more program (Statistica) for hypothesis testing. Now it is verified on two independent platforms.

5.     Authors should provide a better results discussion. focusing on how they can be interpreted comparing to previous studies. In fact, the discussion section corresponds to a set of general ideas, and a presentation of few previous studies, conducted in other contexts, but without a real comparison/discussion of results.

We tried to add more sources with similar research for comparison and completed the discussion. However, the problem is, that there is no similar research done in the Czech Republic. We are the first ones dealing with this topic. At least, we did the comparison with other countries.

6.     The Conclusions section is fact a repetition/summarizing of results, failing in explaining the different contributions of the study.

We explained the contributions in the conclusion.

Thank you and I wish you a happy new year 2019!

Best wishes,

on behalf of our team

Sylvie Formánková

Reviewer 2 Report

The article is an interesting source of information on the awareness of the young generation about corporate social responsibility. In order to improve the content and form, I propose:

L 61 – invoke for specific examples related to corporate social responsibility and sustainable development in controversial sectors such as mining:

Pactwa et al. Sustainable mining – Challenge of Polish mines, Resources Policy https://doi.org/10.1016/j.resourpol.2018.09.009

Woźniak and Pactwa Responsible Mining—The Impact of the Mining Industry in Poland on the Quality of Atmospheric Air Sustainability 2018, 10(4), 1184; https://doi.org/10.3390/su10041184

L 136 “The research is focused specifically on students of economically oriented higher education institutions” - It should be expected that students of economics or management are better oriented in CSR / SRI issues. Have there been or will be research conducted on students of other fields, e.g. technical?

L 157 H 0 : “Students of economically oriented high education institutions are familiar with the term CSR and know exactly what it means.”-  Everything depends on the teaching programme. Depending on what stage of education, issues concerning CSR appear. What the results confirm.

L 213 - No reference in the text to Table 1. Comment please.

L 270 “56% of Millennials were likely to refuse to work for a company after learning that it was not socially or environmentally responsible (Cone 2006 in [42])”. – the research was conducted in the United States, can the declaration be easily transferred in the case of the Czech Republic?

I also suggest to refer to a direct source, especially when it is available http://www.centerforgiving.org/Portals/0/2006%20Cone%20Millennial%20Cause%20Study.pdf

Table 3 - as in the case of table 1 there is no reference in the content of the article

In addition, please check references to literature, for example L 116 and 117, L 224.

Author Response

Dear Reviewer,

Thank you for your comments. We accepted all of them and reacted according to your requirements.

The article is an interesting source of information on the awareness of the young generation about corporate social responsibility. In order to improve the content and form, I propose:

L 61 – invoke for specific examples related to corporate social responsibility and sustainable development in controversial sectors such as mining:

Pactwa et al. Sustainable mining – Challenge of Polish mines, Resources Policy https://doi.org/10.1016/j.resourpol.2018.09.009

Woźniak and Pactwa Responsible Mining—The Impact of the Mining Industry in Poland on the Quality of Atmospheric Air Sustainability 2018, 10(4), 1184; https://doi.org/10.3390/su10041184

Accepted and completed

L 136 “The research is focused specifically on students of economically oriented higher education institutions” - It should be expected that students of economics or management are better oriented in CSR / SRI issues. Have there been or will be research conducted on students of other fields, e.g. technical?

It will be extended in the future – we started with the group of students of economically oriented studies (Economics and management) where we expect they are later in executive/managerial positions. We added this information in the text.

L 157 H 0 : “Students of economically oriented high education institutions are familiar with the term CSR and know exactly what it means.”-  Everything depends on the teaching programme. Depending on what stage of education, issues concerning CSR appear. What the results confirm.

We explained in the text

L 213 - No reference in the text to Table 1. Comment please.

Completed

L 270 “56% of Millennials were likely to refuse to work for a company after learning that it was not socially or environmentally responsible (Cone 2006 in [42])”. – the research was conducted in the United States, can the declaration be easily transferred in the case of the Czech Republic?

Added other sources proving this fact in CR and other states too.

I also suggest to refer to a direct source, especially when it is available http://www.centerforgiving.org/Portals/0/2006%20Cone%20Millennial%20Cause%20Study.pdf

Done

Table 3 - as in the case of table 1 there is no reference in the content of the article

Completed

In addition, please check references to literature, for example L 116 and 117, L 224.

The sources in line 116 and 117 are original sources mentioned in the paper we used. They are in quotation marks, which means we cite the sentence just as it is stated in the paper. L224 checked.

Thank you and we wish you a happy new year 2019!

Best wishes

On behalf of our team

Sylvie Formánková

Reviewer 3 Report

The article is broadly consistent with the scope of the journal and raises current issues related to CSR. However, in order to increase the number of potential readers, I encourage you to introduce a few additions (articles related to the topic of CSR) and changes (add new information).

1. Line 42 There is no definition CSR of the European Union, https://eur-lex.europa.eu/legal-content/EN/TXT/HTML/?uri=CELEX:52002AE0355&from=EN

2. Line 61 (…) economy, such as the mining industry https://www.mdpi.com/2071-1050/9/11/1903 (please, add the article)

3. The article refers to Czech Republic conditions, however, I have the first important consideration. In order to increase the number of future receipts, please attach the questionnaire as an attachment (no.1) to the article. I appreciate the provided survey link (148 lines), but although it is in English, requires registration, which introduces a significant limitation for international recipients.

4. …(line 272) in [42]) (missing dots). CSR is key here for both enhancing the view of an organization by its current employees,  [in this place, please add a quotation https://www.sciencedirect.com/science/article/pii/S0301420717300508], as (…)

5. In the context of SRI, an important binding element is the addition of a review of global trends in functioning stock indices. On the main global stock exchange markets, social and ecological business responsibility indexes have been functioning for years. American Dow Jones Sustainability Indexes (DJSI) and Calvert Social Index (CSI), English / London FTSE4Good Index Series, FTSE JSE Johannesburg Stock Exchange Responsible Investment Index (RPA), Brazilian Sao Paolo Stock Exchange Corporate Sustainability Index (ISE), Indian Standard & Poor's India Environmental, Social and Governance Index (S & P ESG). In Poland, the RESPECT Index is listed - as the first in Central and Eastern Europe [please cit https://www.sciencedirect.com/science/article/pii/S0301420717300508]. In addition, the global index referring to the reach of North America, Europe, Asia and the Pacific - KLD Global Sustainability Index Series (GSI). How does it look in the Czech Republic, besides SRI funds (Line 83/84) ?

6. Materials and Methods - The hypothesis was tested using the Plotly library used for the multilevel program language Python available here: https://plot.ly/python/t-test/.  Is it possible to provide a link to your data - more details? Thus presenting the results I am not able to verify your correctness of calculations. The Student’s test can be generated in excel (Microsoft Office). Was this application used for comparison purposes?

7. Verify links to tables, drawings and literature, please.

After supplement the article, I apply it for a publication in Sustainability journal.

Finally I wish good luck with the paper and this field of research

Regards

Author Response

Dear Reviewer,

thank you for your comments. We accepted all of them and reacted according to your recommendations:

The article is broadly consistent with the scope of the journal and raises current issues related to CSR. However, in order to increase the number of potential readers, I encourage you to introduce a few additions (articles related to the topic of CSR) and changes (add new information).

Added articles reflecting this issue.

1. Line 42 There is no definition CSR of the European Union, https://eur-lex.europa.eu/legal-content/EN/TXT/HTML/?uri=CELEX:52002AE0355&from=EN

There is a definition by the European Commission - source added.

2. Line 61 (…) economy, such as the mining industry https://www.mdpi.com/2071-1050/9/11/1903 (please, add the article)

Article (and some more) added.

3. The article refers to Czech Republic conditions, however, I have the first important consideration. In order to increase the number of future receipts, please attach the questionnaire as an attachment (no.1) to the article. I appreciate the provided survey link (148 lines), but although it is in English, requires registration, which introduces a significant limitation forinternational recipients.

Questionnaire attached. However, it was part of a large questionnaire which takes an hour to fill in. We´ve chosen only the part we used in this paper.

4. …(line 272) in [42]) (missing dots). CSR is key here for both enhancing the view of an organization by its current employees,  [in this place, please add a quotation https://www.sciencedirect.com/science/article/pii/S0301420717300508], as (…)

We added the quotation.

5. In the context of SRI, an important binding element is the addition of a review of global trends in functioning stock indices. On the main global stock exchange markets, social and ecological business responsibility indexes have been functioning for years. American Dow Jones Sustainability Indexes (DJSI) and Calvert Social Index (CSI), English / London FTSE4Good Index Series, FTSE JSE Johannesburg Stock Exchange Responsible Investment Index (RPA), Brazilian Sao Paolo Stock Exchange Corporate Sustainability Index (ISE), Indian Standard & Poor's India Environmental, Social and Governance Index (S & P ESG). In Poland, the RESPECT Index is listed - as the first in Central and Eastern Europe [please cit https://www.sciencedirect.com/science/article/pii/S0301420717300508]. In addition, the global index referring to the reach of North America, Europe, Asia and the Pacific - KLD Global Sustainability Index Series (GSI). How does it look in the Czech Republic, besides SRI funds (Line 83/84) ?

It was described.

6. Materials and Methods - The hypothesis was tested using the Plotly library used for the multilevel program language Python available here: https://plot.ly/python/t-test/.  Is it possible to provide a link to your data - more details? Thus presenting the results I am not able to verify your correctness of calculations. The Student’s test can be generated in excel (Microsoft Office). Was this application used for comparison purposes?

I can share the documents with you: https://uloz.to/tam/_gj8BzCAsK03e

However, most of the data is in Czech language. We tested the results with another porgram (Statistica). Results were the same. So now it is verified on two independent platforms. If you need anything else, let me know.

7. Verify links to tables, drawings and literature, please.

Links verified.

Thank you very much and wish you a happy new year 2019!

Best wishes

Sylvie Formánková

Round  2

Reviewer 1 Report

Review Report

As I referred previously, I firmly believe that there is a real need to conduct researches focusing Millennials’ attitudes and behaviours towards CSR issues, addressing thus a significant gap in the CSR literature. Yet, unfortunately, I still believed that, although in general the study is well presented, I don’t think that the paper rises to the standards expected by the Sustainability journal. Indeed I still have several concerns and questions that, as a reviewer I would like to bring up:

1.      Overall, it is still unclear what is really new in this paper from a CSR theory or managerial practice perspective. Indeed, several similar studies (and inclusive much more developed) have already been conducted around the world. It is true that authors have rephrased their objective to “The aim of this paper is to examine the awareness of the Czech Millennials of corporate social responsibility (CSR) and their attitude to sustainable and responsible investment (SRI).”, focussing clearly on the Czech context. In accordance, and considering the number of studies already published about other geographical contexts, authors still fail to show a significant gap related to the studies examining Millennials’ attitudes and behaviours towards CSR issues, and consequently in justifying properly the paper’s purpose. The introduction still fails clearly in explaining the real significance of the study. Why is it important to study and report the Czech context, in addition to so many studies already published? For whom is it important? Authors should justify that this is not “one more” study.

2.      As suggested previously, authors did split the former introduction into two different sections: Introduction and Theoretical Background. However, the new Theoretical Background section remains under-developed. Considering the main objectives of the paper, readers would expect in this section, a review concerning factors that may influence students in general, and millennials in particular, regarding their perception about CSR issues (historical context, cultural context, political context, …). Such a theoretical support would help in discussing results in the “Discussion section”.

3.      If authors pretend to keep the hypotheses, these should be better sustained (grounded in previous literature), as previously highlighted. Otherwise, probably it would be easier to forget the hypotheses and concentrate on the different research questions.

4.      Authors did try to improve the discussion section. However, authors still need to provide a better results discussion. focusing on how they can be interpreted comparing to previous studies. In fact, the discussion section still corresponds to a set of general ideas, and a presentation of few previous studies, conducted in other contexts, but without a real comparison/discussion of results. Authors refer that “In terms of the Millennials attitude to sustainable investing it is very difficult to compare the data. There haven´t been a common research conducted in the Czech Republic.”. The lack of research in the Czech context is irrelevant here, because the idea would be to compare authors’ findings with other researchers’ findings in other contexts, and discuss similarities/differences in order to legitimize some kind of contribution to theory/practice.

5.      The Conclusions section was not improved, remaining a repetition/summarizing of results, failing in explaining the different contributions of the study.

Author Response

Dear reviewer,

thank you for your comments and recommendations. We really appreciate it and believe it helps to improve the quality of our paper. Based on the recommendation of the two other reviewers we deleted the first table. They didn´t have any other comments and recommended the article for publishing. 

We did all our best to correct the article according to your comments and recommendations. We added some comments in the paper - please, check the text. We reacted to all your recommendations:

1.      Overall, it is still unclear what is really new in this paper from a CSR theory or managerial practice perspective. Indeed, several similar studies (and inclusive much more developed) have already been conducted around the world. It is true that authors have rephrased their objective to “The aim of this paper is to examine the awareness of the Czech Millennials of corporate social responsibility (CSR) and their attitude to sustainable and responsible investment (SRI).”, focussing clearly on the Czech context. In accordance, and considering the number of studies already published about other geographical contexts, authors still fail to show a significant gap related to the studies examining Millennials’ attitudes and behaviours towards CSR issues, and consequently in justifying properly the paper’s purpose. The introduction still fails clearly in explaining the real significance of the study. Why is it important to study and report the Czech context, in addition to so many studies already published? For whom is it important? Authors should justify that this is not “one more” study.

We tried to clearly describe the gap and show the lack of the scientific papers dealing with this issue and we tried to point the importance of solving this issue in the part of the introduction. We added new articles, did a complete analysis of all articles dealing with the same or similar topic at Web of Science database. We identified the stakeholders for those is the article important. We tried to justyfy that it is not "one more" study. I believe it is obvious now.

2.      As suggested previously, authors did split the former introduction into two different sections: Introduction and Theoretical Background. However, the new Theoretical Background section remains under-developed. Considering the main objectives of the paper, readers would expect in this section, a review concerning factors that may influence students in general, and millennials in particular, regarding their perception about CSR issues (historical context, cultural context, political context, …). Such a theoretical support would help in discussing results in the “Discussion section”.

We added other articles, books and literature sources explaining the factors influencing the millennials, tried to explain it from the historical, cultural context...however making a complete discussion and review it would be for another paper...many sources, many different attitudes etc. We pointed those that are relevant to our issue.

3.      If authors pretend to keep the hypotheses, these should be better sustained (grounded in previous literature), as previously highlighted. Otherwise, probably it would be easier to forget the hypotheses and concentrate on the different research questions.

We kept the hypothesism we consider it important. We added papers confirming our premises. 

4.      Authors did try to improve the discussion section. However, authors still need to provide a better results discussion. focusing on how they can be interpreted comparing to previous studies. In fact, the discussion section still corresponds to a set of general ideas, and a presentation of few previous studies, conducted in other contexts, but without a real comparison/discussion of results. Authors refer that “In terms of the Millennials attitude to sustainable investing it is very difficult to compare the data. There haven´t been a common research conducted in the Czech Republic.”. The lack of research in the Czech context is irrelevant here, because the idea would be to compare authors’ findings with other researchers’ findings in other contexts, and discuss similarities/differences in order to legitimize some kind of contribution to theory/practice.

We added new papers, however we believe there is sufficient discussion of our results with other researchers from different countries. Nevertheless, as we pointed in the introduction part, in the last 5 years there aren´t many papers dealing with similar or relevant issues to ours. Inspite of this fact we added also older approaches in the discussion part.

5.      The Conclusions section was not improved, remaining a repetition/summarizing of results, failing in explaining the different contributions of the study.

We pointed the importance of our article and the contribution of our study.

We strongly believe the paper is already acceptable for publishing since we really worked hard to improve it according to all comments and recommendations. The attitudes how to approach such topic might differ. We believe you accept our point of view. 

Thank you very much.

Best regards,

Sylvie Formánková

Reviewer 2 Report

Dear Authors

I inserted corrections in the attached file. After their introduction, I give a positive opinion.
I wish you happy new year.

Regards

Author Response

Dear reviewer,

thank you for your comments and recommendation. We deleted the table and made other corrections recommended by other reviewers.

Best regards,

Sylvie Formánková

Reviewer 3 Report

Dear Authors

Thank you for the improvement according to my comments. After re-reading the article, please delete Table No. 1, because Figure No. 1 is enough. The information is duplicated in the text and the detailed reference is to Figure 1.

Good luck in further research Regards

Author Response

Dear reviewer,

thank you for your comments and recommendation. We deleted the table and made other corrections recommended by the other reviewers.

Best regards,

Sylvie Formánková

Round  3

Reviewer 1 Report

The revision submitted has improved a lot the paper’s quality. Authors endeavoured to improve the paper according to the different comments and recommendations provided in the previous revision process.

Authors identified stakeholders for those the article is important, improved the theoretical background, which obviously also helped in improving the discussion section.

Overall, this last revision helped to improve the quality of the paper, and the importance of the paper and contributions of the study are now more clear to readers.

In accordance, my opinion is that the paper is now acceptable for publishing.

Best regards